# Critical Periods, Critical Time Points and Day-of-the-Week Effects in COVID-19 Surveillance Data: An Example in Middlesex County, Massachusetts, USA

**DOI:** 10.3390/ijerph19031321

**Published:** 2022-01-25

**Authors:** Ryan B. Simpson, Brianna N. Lauren, Kees H. Schipper, James C. McCann, Maia C. Tarnas, Elena N. Naumova

**Affiliations:** Division of Nutrition Epidemiology and Data Science, Friedman School of Nutrition Science and Policy, Tufts University, 150 Harrison Avenue, Boston, MA 02111, USA; ryan.simpson@tufts.edu (R.B.S.); brianna.lauren@tufts.edu (B.N.L.); kschipper0430@comcast.net (K.H.S.); james.mccann@tufts.edu (J.C.M.); maia.tarnas@gmail.com (M.C.T.)

**Keywords:** COVID-19, data reliability, day-of-the-week effects, Kolmogorov-Zurbenko filters, nonpharmaceutical interventions, precision public health, stringency

## Abstract

Critical temporal changes such as weekly fluctuations in surveillance systems often reflect changes in laboratory testing capacity, access to testing or healthcare facilities, or testing preferences. Many studies have noted but few have described day-of-the-week (DoW) effects in SARS-CoV-2 surveillance over the major waves of the novel coronavirus 2019 pandemic (COVID-19). We examined DoW effects by non-pharmaceutical intervention phases adjusting for wave-specific signatures using the John Hopkins University’s (JHU’s) Center for Systems Science and Engineering (CSSE) COVID-19 data repository from 2 March 2020 through 7 November 2021 in Middlesex County, Massachusetts, USA. We cross-referenced JHU’s data with Massachusetts Department of Public Health (MDPH) COVID-19 records to reconcile inconsistent reporting. We created a calendar of statewide non-pharmaceutical intervention phases and defined the critical periods and timepoints of outbreak signatures for reported tests, cases, and deaths using Kolmogorov-Zurbenko adaptive filters. We determined that daily death counts had no DoW effects; tests were twice as likely to be reported on weekdays than weekends with decreasing effect sizes across intervention phases. Cases were also twice as likely to be reported on Tuesdays-Fridays (RR = 1.90–2.69 [95%CI: 1.38–4.08]) in the most stringent phases and half as likely to be reported on Mondays and Tuesdays (RR = 0.51–0.93 [0.44, 0.97]) in less stringent phases compared to Sundays; indicating temporal changes in laboratory testing practices and use of healthcare facilities. Understanding the DoW effects in daily surveillance records is valuable to better anticipate fluctuations in SARS-CoV-2 testing and manage appropriate workflow. We encourage health authorities to establish standardized reporting protocols.

## 1. Introduction

One of the most important lessons of the novel coronavirus 2019 pandemic (COVID-19) has been the importance and utility of real-time infectious disease surveillance for monitoring, tracking, and reducing emerging infectious outbreaks [1,2,3]. Worldwide, and in the United States, SARS-CoV-2 surveillance has been made possible by collaborations between governmental agencies, public health authorities, healthcare systems, academic institutions, and media outlets to collect, curate, distribute, and communicate spatially and temporally refined health surveillance records [4,5]. Unlike most publicly available infectious disease surveillance data, SARS-CoV-2 reported tests, cases, and deaths have been curated at a daily temporal resolution and state- and county-level geographic areas. These reporting protocols create unique opportunities for public health professionals to explore the patterns associated with broad intervention measures or often common yet overlooked patterns in reported data. The systematic changes in COVID-19 outcomes triggered by various interventions are under ongoing investigations by many research groups, yet a more granular and less obvious weekly variations are less explored.

One such pattern is day-of-the-week (DoW) effects, which describe differences in the reporting of health outcomes for some weekdays compared to a single reference day. To date, few studies have aimed to describe and quantify DoW effects in daily SARS-CoV-2 surveillance data. Instead, many studies aggregate or average daily records into weekly counts or rates or adjust for the DoW effects with various statistical techniques and thus ignore the potentially substantial and informative variations and differences between weekday or weekend reporting or their changes over time [6,7,8,9]. However, DoW effects reflect important patterns in the availability and capacity of laboratory testing facilities as well as popular preferences for seeking testing. The presence of DoW effects help to inform public health testing facilities and laboratories which days of the week have the highest or lowest demands for testing. In doing so, these facilities and public health agencies can better collaborate, anticipate, and manage laboratory testing resources and the scheduling of personnel to accommodate higher testing demands [10,11,12,13]. By examining how DoW effect sizes change over time, public health professionals can also monitor changes in popular preferences throughout the year, such as before and during times of federal or religious holidays or after the relaxation of non-pharmaceutical interventions [14,15,16,17]. This real-time surveillance improves the timeliness of test stockpiling and improves testing availability using mobile testing, the opening of additional test clinics, or subsidizing at-home, low-cost testing kits.

Examination of DoW effects has proven extremely beneficial in public health planning and programming for other respiratory illnesses and burdens. For example, influenza and influenza-like-illnesses have demonstrated clear increases in the reporting of tests and cases on weekdays compared to weekends due to laboratory testing capacity [18,19]. These findings have informed patterns of seasonal vaccination rollout and preparations for increased influenza testing capacity to accommodate increased volumes of seasonal influenza cases. Other calendar effects, including holiday effects, have identified amplification and dampening of influenza incidence due to social holidays and school closures, respectively [19,20,21]. Within environmental health sciences, DoW effects have been recognized for a long time in both the exposure and outcome variables [22,23].

One of the challenges of studying DoW effects is the need for assuring data quality, the absence of structural missingness or spikes (say during weekends or holidays), and consistency in reporting of time-referenced records. Now almost 2 years into the ongoing pandemic, several reports and commentaries have discussed limitations of publicly available COVID-19 surveillance data and the need for improvement as the pandemic persists [24,25,26,27,28,29,30,31]. For example, many statewide agencies use a variety of dates for reported cases such that a positive test result could be reported by the date of symptom onset, specimen collection, positively identified culture, or date of reporting [27,31]. Similarly, the definition of SARS-CoV-2 deaths may rely on a positive test shortly before or after death (which would underestimate deaths depending on testing availability) or include individuals who were not tested but were suspected of having SARS-CoV-2 (likely overestimating deaths) [29]. Despite these criticisms, few studies have quantified patterns to reporting inconsistencies or missing data in commonly used publicly available surveillance data [32,33].

In this study, we examined DoW effects using publicly available SARS-CoV-2 surveillance data reported from 2 March 2020 through 7 November 2021. We used John Hopkins University’s (JHU’s) Center for Systems Science and Engineering (CSSE) COVID-19 data repository in Middlesex County, Massachusetts, USA. We inspected and cross-referenced JHU data with the records of the Massachusetts Department of Public Health (MDPH) COVID-19 data dashboard. We defined major waves, their critical periods and points—constituting so-called outbreak signatures—for reported tests, cases, and deaths using Kolmogorov-Zurbenko adaptive filters with a 21-day smoothing window. We modeled DoW effects using segmented time series analyses that adjusted for linear and quadratic trends within outbreak signatures’ critical periods. We evaluated differences in DoW effects by Massachusetts non-pharmaceutical intervention phases and their stringency. By exploring DoW effects, public health professionals can identify patterns of reporting and begin to speculate testing capacity limitations and popular testing preferences to precisely model outbreak signatures.

## 2. Methods

### 2.1. Data Extraction and Management

In December 2019, JHU CSSE designed an online interactive dashboard to curate and report cumulative incidence of SARS-CoV-2 cases, deaths, and recoveries [34]. Scraping electronic health records from city, state, and national health agencies, this publicly available data repository provided the most extensive reporting of SARS-CoV-2 information in the United States [35,36]. We extracted cumulative counts of cases and deaths for all US counties from Monday, 2 March 2020 through Sunday, 7 November 2021 [37].

Upon extraction, we subset data for Middlesex County, Massachusetts as a case study to explore DoW effects in publicly available surveillance data. We selected Massachusetts as it ranked 2nd nationwide in healthcare access and 4th nationwide in public health spending throughout our study period [38]. These rankings suggested that Massachusetts would have highly reliable and complete SARS-CoV-2 surveillance data. We selected Middlesex County as it was the most populous county in the state (~25% of residents) and the New England region [39].

JHU metadata reported several modifications in historic records as of 10 November 2021 (data download date) [40]. First, JHU back-distributed probable cases previously reported on 12 June 2020 across 15 April through 11 June 2020 in accordance with revised MDPH records. Second, MDPH revised reporting methodology of cumulative cases and deaths on 19 August 2020. New guidelines transitioned from aggregating probable and confirmed cases to reporting confirmed cases only. Reporting methodology changed again on 3 September 2020 when MDPH revised definitions of probable cases to exclude antibody tests, resulting in the removal of 8051 historic reported cases statewide [40].

Similarly, JHU reported that 680 positive cases prior to 1 December 2020 were not reported by MDPH due to technical issues at laboratory testing facilities [40]. JHU redistributed historic cases, yet no metadata described how such redistribution occurred. Finally, JHU uploaded electronic records from MDPH weekly on Monday-Friday only; metadata provided no information on how weekend records were reported. While JHU reported state and county level test data nationwide from 2 March 2020 through 1 August 2021, no records were available for Middlesex County or Massachusetts [41].

We extracted MDPH cumulative reports for SARS-CoV-2 cases and deaths to validate JHU records [42,43]. MDPH reported cumulative cases starting 17 April 2020 while cumulative deaths were reported starting 1 June 2020. Neither cases nor deaths were reported from 12–18 August 2020, one week prior to changes in statewide case reporting guidelines. We extracted cases and deaths for Middlesex County, however no county-level reported tests were available [42,43]. Instead, we extracted state-level reported tests to approximate DoW effects in Middlesex County.

We estimated daily tests, cases, and deaths by subtracting cumulative counts of health outcomes from successive days. We estimated daily rates as the ratio of daily counts to annual population estimates (drawn from the 2010 US Census) and multiplied by 1,000,000. We reported tests, cases, and deaths per 1,000,000 persons (‘tpm’, ‘cpm’, and ‘dpm’, respectively).

### 2.2. Inspection of Data Reliability

We examined JHU data reliability for health outcomes using 3 metrics. First, we explored quantities and patterns of completeness, defined as the number of time points with non-missing records [44]. Completeness reflected the number of days with usable health records in our study period. We differentiated incomplete records (i.e., blanks) from days with 0 counts and examined patterns of completeness by DoW and Gregorian calendar date.

Next, we examined data precision, defined as the plausibility of case information (e.g., negative counts). Data imprecision suggested inaccurately reported or modified records. We corrected negative counts by replacing negative values with 0 counts and subtracting 1 test, case, or death from as many days as the absolute value of negative counts.

Finally, we explored data interoperability, defined as the concordance between surveillance systems’ records. As JHU extracted data directly from MDPH, we expected strong concordance between systems for cases and deaths. We identified days with concordant and discordant data to clarify discrepancies between surveillance systems.

### 2.3. Defining Outbreak Signatures

By defining outbreak signatures, or the profile of an epidemic curve and its features, modelers can precisely quantify and adjust for complex temporal trends [45,46,47]. Signatures consist of 3 features: (i) rate magnitudes, (ii) critical points (e.g., onset, peak, resolution), and (iii) critical periods (e.g., acceleration, deceleration, steady state). Critical points represent time points when rate magnitudes reach local minima and maxima. These time points define key features of an epidemic curve such as the onset, peak, or resolution of an outbreak wave. Furthermore, these points reflect transitions between waves of persistent outbreaks or primary and secondary peaks within a single outbreak wave. Critical periods, defined as the duration between consecutive critical points, reflect intervals when the pace of rates accelerate, decelerate, or remain unchanged over time. These periods represent well-defined linear and non-linear temporal trends of rates.

We estimated rate magnitudes using a version of Kolmogorov-Zurbenko (KZ) adaptive filters. This smoothing technique delineates trends in noisy time series by generating average estimates across a moving window of time points [48,49,50,51]. We used a 21-day smoother, which ensured sufficient reduction of daily noise without overly aggregating daily records (as with monthly smoothers). Furthermore, this smoother ensured low probability of erroneous critical points (more probable with 7- or 14-day smoothers) and ensured equal weighting of DoW observations compared to non-7-day smoothing windows.

We defined critical points using rate magnitudes and their trivial derivatives, calculated as the difference between rates of successive days. We identified peaks as days with maximum rate magnitudes and low or near-zero trivial derivative values. We identified onset points as days with local minimum rate magnitudes when trivial derivatives began increasing from near-zero to highly positive values, marking the beginning of an outbreak wave. We identified resolution points as days with local minimum rate magnitudes when trivial derivatives declined from highly positive to near-zero or near-minimum values, marking the conclusion of an outbreak wave.

With respect to critical periods, we defined acceleration periods from outbreak onset points to peaks when the pace of rates steadily increased. In contrast, we defined deceleration periods from outbreak peaks to resolution or next wave onset points when the pace of rates steadily decreased. We defined steady state periods as the duration between the resolution of one outbreak wave and the onset of another when rates remain near-constant and trivial derivatives were consistently near-zero values.

Outbreak waves reflected the duration from onset to resolution critical points. For many waves, no resolution occurred; the conclusion of one wave marked the onset of the next. As such, outbreak waves were also defined as the duration between 2 onset points if no resolution occurred. We identified primary and secondary outbreak peaks for specific waves when rates reached 2 local maxima after abrupt changes in the trivial derivative from systematically high negative to systematically high positive values.

### 2.4. Defining Intervention Stringency Phases

Stepwise implementation of statewide non-pharmaceutical interventions intended to improve SARS-CoV-2 testing and reduce disease transmission beginning 15 March 2020 [52]. These interventions included stay-at-home advisories and business closures and reopening restrictions to reduce SARS-CoV-2 transmission. Statewide policies closely aligned with county-level mask mandates, business curfews, lock-down restrictions, and school closures also intending to disrupt SARS-CoV-2 transmission (Appendix A). As case and death counts increased and declined, intervention mandates gradually became more stringent or relaxed their stringency, respectively.

Changes in statewide advisories defined 9 intervention phases beginning 22 January 2020 through 28 May 2021 (Table 1). We scored phases by the stringency of policies for restricting social mobility and reducing disease transmission. Scores of 5 represented most stringent policies (e.g., school closures, stay-at-home order) while scores of 0 represented least stringent policies (e.g., prior to and after the Massachusetts state of emergency). We examined differences in DoW effects by phase, as statewide mandates influenced reporting patterns, laboratory testing capacity, and testing preferences. We also compared the alignment of outbreak signature critical points and intervention phases to better understand delays in health policy implementation.

### 2.5. Modeling DoW Effects

We estimated average (with 95% confidence intervals) and median (with interquartile range) daily rates and product moments by DoW. We selected L-skewness and L-kurtosis given their sensitivity for distributions of low rates and smaller sample size [53]. High values of L-moments indicate a possible surge in tests, cases, or deaths and are necessary for justifying the use of negative binomial distributions when modeling count-based distributions. We estimated average rates using generalized linear models adjusted for a negative binomial distribution and log-link function to accommodate moderately skewed count-based data:(1)$$\ln\left[E\left(Y_{j}\right)\right]=\beta_{1}(D_{t})$$
where *Y_j,d_*—daily rates of *j*-outcome (e.g., tests, cases, deaths) for *d*-DoW. We estimated average and confidence interval values by exponentiating model coefficients ($\exp\left(\beta_{0}\right)$ and $\exp\left(\beta_{0}\pm 1.96se\right)$, respectively).

Next, we compared DoW effects adjusting for linear (*t*) and quadratic (*t*^2^) trends in outbreak signature critical periods. This adjustment captured general increases and decreases in rate magnitudes as well as acceleration and deceleration in the pace of rates. We used a segmented negative binomial regression model, such that:(2)$$\ln\left[E\left(Y_{j,t}\right)\right]=\beta_{0}+\beta_{1}(D_{t})+\overset{q}{\underset{k=1}{\sum }}\left[\beta_{2k}\left(t-t_{k}\right)+\beta_{2k+1}{\left(t-t_{k}\right)}^{2}\right]\;$$
where *Y_j,t_*—daily rates for *j*-outcome on *t*-day; *D_t_*—categorical variable for DoW on *t*-day (Reference: Sunday); and (*t* − *t_k_*)—continuous time series variables (in days) for *k*-critical period of *j*-outcome. Here, *q* corresponded to the total number of critical periods for tests (10), cases (8), and deaths (8), as defined by our study period and KZ adaptive filters.

We estimated DoW effects for each intervention phase using a similar model:(3)$$\ln\left[E\left(Y_{j,t,p}\right)\right]=\beta_{0}+\beta_{1}(D_{t,p})+\overset{q}{\underset{k=r}{\sum }}\left[\beta_{2k}\left(t-t_{k}\right)+\beta_{2k+1}{\left(t-t_{k}\right)}^{2}\right]$$
where *p*—policy intervention phase ranging from A–I as reported in Table 1. Here, *r* corresponded to the first outbreak critical period occurring within *p*-intervention phase.

We compared model fit using Akaike’s Information Criterion (AIC) estimates. We defined statistical significance as α < 0.05. We performed data extraction, alignment, management, and cleaning using Excel 2016 Version 2103 and R (4.0.0) software. We conducted statistical analyses and created data visualizations using SE/16.1 and R (4.0.0) software. We share all R software codes as Appendix A.

## 3. Results

### 3.1. JHU and MDPH Data Cross-Referensing

While JHU data had 100% completeness, we found several discrepancies in daily counts including 3 days with negative values in cases (3 September 2020, 2 March 2021, and 27 June 2021) and 4 days with negative values in deaths (30 June 2020, 21 July 2020, 2 September 2020, and 9 July 2021). Negative values ranged from −16 to −69 for cases and −1 to −11 for deaths. Negative values in September 2020 coincided with revisions in MDPH guidelines to exclude probable cases with positive antibody tests and were found in JHU data only. Negative values for cases and deaths in March–July of 2020 and 2021 were reported by both JHU and MDPH. We detected no negative values for state-level tests.

JHU systematically reported 0 (zero) counts for cases and deaths on federal and religious holidays. These included Thanksgiving (26 November 2020), Christmas (25 December 2020), New Year’s (1 January 2021), Easter (4 April 2021), Memorial Day (31 May 2021), Independence Day (5 July 2021), Labor Day (6 September 2021), and Columbus/Indigenous People’s Day (11 October 2021). Similarly, JHU systematically reported 0 counts on all weekend days beginning 3 July 2021. When compared to MDPH, we found incomplete records (blanks) on these holidays and weekends suggesting no statewide reporting. Thus, JHU reported 0 counts on these holidays and weekends as cumulative counts remained unchanged in the absence of MDPH reporting.

We assessed the interoperability between JHU and MDPH beginning 17 April 2020 for cases and 1 June 2020 for deaths. JHU case records were systematically discordant with MDPH records from 17 April 2020 through 3 September 2020. On 3 September 2020, JHU reported negative counts, which reflected the correction of cumulative counts needed to achieve the concordance of JHU records with MDPH records for the remainder of our study period. The death records for JHU and MDPH were the same for the study period.

### 3.2. Outbreak Signatures and Their Features

We identified 5 distinct outbreak waves for statewide reported tests with the highest peak values on 6 December 2020 (Wave 3; 3213.94 tpm) and 15 September 2021 (Wave 5; 4412.13 tpm) (Table 2 and Table 3, Figure 1). Wave 3 peak timing aligned with the onset of non-pharmaceutical intervention Phase F, which returned to more stringent health mandates after an outbreak in reported cases. Tests reached a global minimum at Wave 5 onset (27 June 2021; 886.56 tpm), with rate magnitudes paralleling Wave 2 onset a year prior (4 June 2021; 1171.02 tpm). Wave 3 onset (25 September 2021) aligned closely with intervention Phase E (29 September 2020), which relaxed the stringency of health mandates most compared to prior phases (Figure 2). Wave 5 onset came ~1 month after lifting the Massachusetts State of Emergency (Phase I; 28 May 2021) and peaked on 15 September 2021.

We identified a global maximum of cases for the primary peak of Wave 2 (2 January 2021; 591.37 cpm), which trailed peak tests and a return to more-stringent intervention Phase F by ~1 month. We also identified a Wave 2 secondary peak on 1 April 2020 (242.55 cpm), which trailed the relaxation to less-stringent intervention Phases G and H (25 February 2021 and 18 March 2021, respectively) by ~1 month. Rate magnitudes of the Wave 2 secondary peak paralleled Wave 1 peak (264.01 cpm). Similarly, Wave 3 onset (23 June 2021; 9.80 cpm) returned to similar magnitudes as Wave 2 onset (4 July 2020; 21.29 cpm) ~1 year later. Rates of reported cases reached steady state at ~165 cpm after peaking for Wave 3 on 12 September 2021, which aligned with the global maxima in reported tests.

Wave 1 peak had the highest reported rates of death (22.51 dpm), which were ~3-times higher than Wave 2 peak rate magnitudes (8.00 dpm). Peaks in reported deaths (30 April 2020, 15 January 2021, 1 October) trailed peaks in reported cases (20 April 2020, 2 January 2021, 12 September 2021) by ~10–20 days. Resolution critical points of similar magnitudes for reported deaths were observed in Wave 1 (8 August 2020; 1.72 dpm) and Wave 2 (7 July 2021; 0.30 dpm). Like with reported cases, these resolution critical points separated consecutive outbreak waves by ~1 year.

### 3.3. Day-of-the-Week Effects

The estimated median values for reported tests, cases, and deaths along with their confidence intervals and coefficients of skewness and kurtosis are shown in Table 4. As expected for each death there were about 30 cases of infections, and for each reported case there were about 10 reported tests. Median rates were high on weekdays for tests and cases yet low on weekends; we found little differences between mean or median rates of deaths across all days of the week. Relatively high values of L-skewness and L-kurtosis for daily death cases suggested the potential for high temporal variability.

For the entire study period, reported tests were ~2.00–2.30-times more likely to be reported on weekdays compared to Sunday (Table 5). For all weekdays, the size of this effect decreased consistently from more stringent Phase C (RR = 2.45–2.88 [2.01, 3.53]) to less stringent Phase G (RR = 1.72–2.24 [1.63, 2.35]). Effect sizes slightly increased for Tuesdays-Thursdays in Phases H–I, which aligned with the global maxima of reported tests in outbreak Wave 5. In contrast to weekdays, tests were only 1.26-times [1.17, 1.35] more likely to be reported on Saturday compared to Sunday. We found lower effect sizes for reported tests on Saturday compared to Sunday in least stringent Phases G (RR = 1.19 [1.13, 1.25]) and H (RR = 1.19 [1.12, 1.27]). In the last Phase I, there were substantial drops in testing on weekends and no significant differences in reported tests on Saturday compared to Sunday. Analyses for individual intervention phases largely support the findings observed for the entire study period (Figure 3).

Like statewide tests, county-level cases were ~1.75–2.00-times more likely to be reported on weekdays compared to Sunday; cases reported on Saturday were only 1.32-times [1.04, 1.67] higher as compared to Sunday. For all weekdays, the effect size decreased consistently from more stringent Phase B (RR = 2.06–2.25 [1.81, 2.56]) to less stringent Phase D (RR = 1.53–1.87 [1.13, 2.54]). However, DoW effects changed substantially in less stringent Phases E–H, which aligned with the overall testing and case reporting. Furthermore, cases were significantly less likely to be reported on Mondays (RR: 0.66–0.93 [0.47, 0.97]) in Phases E–G and Tuesdays (RR: 0.51–0.85 [0.44, 0.88]) in Phases F–G during the large second wave.

Overall, we found minimal or no significant differences in reported deaths between any days of the week, especially in the most stringent Phases B–C. As later intervention phases had many weeks without reported deaths, confidence intervals were exaggerated in regression analyses. The observed increase in reported deaths on Wednesdays and Thursdays (1.46-times [1.15, 1.85], *p* = 0.002) and 1.28-times ([1.00, 1.64], *p* = 0.047, respectively) were driven by spikes in deaths in individual critical periods and intervention phases.

## 4. Discussion

The presented investigation described and compared patterns in the reporting of publicly available statewide SARS-CoV-2 tests and Middlesex County reported cases and deaths from 2 March 2020 through 7 November 2021. First, we identified distinct waves with their critical periods using a novel application of Kolmogorov-Zurbenko adaptive filters that allowed us to define, describe, and compare features of outbreak signatures. We found patterns in the timing of wave onset, peak, and resolution critical points compared to non-pharmaceutical intervention phases. Given such characterization, we identified significant DoW effects for Massachusetts statewide tests and Middlesex County cases with distinct patterns across intervention phases. We also identified and reconciled systematic reporting inconsistencies between JHU and MDPH surveillance systems, including dissimilar missing data reporting protocols and the reporting of negative counts to correct historic records. These findings provide critical insight on the systematic features in daily patterns of major pandemic outcomes, such as tests, cases, and deaths. These findings help to understand the utilization of laboratory capacity and popular preferences of testing that can assist public health agencies in effectively distributing testing resources and managing laboratory personnel. The combination of these findings indicates health professionals must place greater attention to investigating DoW effects and other reporting inconsistencies in public surveillance data.

In Massachusetts, the signature of statewide reported tests aligned closely with the transition between non-pharmaceutical intervention phases. Wave 3 onset timing (25 September 2021) aligned with the relaxation to less-stringent Phase E (29 September 2020) while Wave 3 peak timing (6 December 2020) aligned with the return to more-stringent Phase F (8 December 2020). Similarly, Wave 4 onset timing (10 March 2021) aligned closely with the relaxation to less-stringent Phase H (18 March 2021). The acceleration of reported tests after phase stringency relaxation reflects increased anxiety of disease transmission with greater social mobility or necessity for pooled testing in schools or workplaces and inpatient testing even among those without presenting symptoms [54,55,56,57,58,59].

Alternatively, increased testing may reflect premature relaxation in the stringency of non-pharmaceutical intervention phases. We found that the onset timing of reported cases in Wave 2 (4 July 2020 and 2 March 2021) aligned closely with the relaxation to intervention Phase D (2 July 2020) and Phase G (25 February 2021). We also found that the peak timing of reported deaths trailed the peak timing of reported cases by ~10–20 days. Thus, increases in social mobility and business/workplace re-openings resulted in almost immediate increases in reported cases of high morbidity/mortality [60,61].

Shared signatures of reported tests and cases also reflect potential seasonal patterns of testing behaviors. Tests peak timing in December 2020 preceded cases peak timing in January 2021 by ~1 month. Yet, both health outcomes peaked at nearly the same time in April and September 2021. These changes reflect increased testing volumes in anticipation of seasonal travel in December as well as increased cases after wintertime social mixing [62,63,64,65,66]. Additionally, reported tests and cases reached near-identical rate magnitudes in June/July 2020 as June/July 2021 after their wintertime peaks. These signatures resemble seasonal patterns of influenza, which are often associated with patterns of school openings and increased social mixing [20,67,68]. While research already suggests an underlying endemic seasonality of SARS-CoV-2 [69,70], no definitive seasonal pattern can be determined without further research across a longer time series length.

DoW effects in reported tests reflect a combination of laboratory testing capacity and popular testing preferences. Tests were more likely to be reported on weekdays than on Saturday or Sunday. These patterns reflect the limited testing and reporting capacity, particularly at the beginning of the COVID-19 pandemic [71,72]. Greater DoW effects across weekdays also reflects greater required or preferred testing during weekdays. Furthermore, we found changes in testing across intervention phases that likely reflect changes in health safety precautions throughout the pandemic, such as preparation for weekend social activities and travel [73] or mandates for testing in workplaces, schools, and other transmission-prone environments [74,75].

We found similar patterns in county-level reported cases as with statewide tests. Like tests, DoW effect sizes for reported cases decreased consistently from more stringent Phase B to less stringent Phase D. Changes in weekday DoW effects for reported cases might reflect changes in hospitalization patterns, severity of infections, and potential delays due to high testing volumes earlier in the week or during wave peaks. DoW effects for tests and cases may also reflect changes in transmission and social mixing when stay-at-home policies and business and retailer closures were relaxed in 2021. Alternatively, we found no DoW effects for deaths, as the likelihood of death was equally probable on any day of the week. Yet, we found several spikes in death counts we can’t explain; those spikes triggered occasional significant effects. These spikes are supported by high values of L-skewness and L-kurtosis suggesting temporal variability of deaths within our time series. Future studies must examine compound effects of additional contributors, like local air quality [76], extreme weather [77], specific calendar events (see Appendix A), an introduction of rapid over-the-counter tests [11] influencing viral transmission and thus, daily COVID-19 outcomes. Studies may also consider more exhaustive adjustment of confounding factors expected to influence case or death onset such as age, chronic health conditions, body mass index, etc.

Our findings demonstrate serious concerns over how longitudinal public health surveillance death records are curated and reported. This largely stems from discrepancies in how dates are reported, such as the date of death or the date of reporting SARS-CoV-2 as the cause of death, and potential weekly batch reporting rather than consistent reporting throughout every week [78]. These reporting patterns question the reliability and usability of SARS-CoV-2 death records for examining temporal patterns in case severity. We encourage the public health community to incorporate how test, case, and death dates are reported in standardized case definitions of infectious disease health outcomes.

One of the major advantages of publicly available SARS-CoV-2 surveillance data are their temporal and spatial granularity that allow not only a more precise estimation of outbreak signatures but also highlights the need for improved standardization of reporting protocols for publicly available infectious disease surveillance data. We found that both JHU and MDPH reports of cumulative counts produce biologically implausible negative values that aligned closely with historical record modifications in September 2020 [40]. After negative value corrections on 3 September 2020, JHU reported cases were concordant with MDPH records. While metadata report the presence of historic changes, neither database thoroughly explains techniques used to make these corrections. The appearance of negative values creates well justified distrust, which can lead data users to question the reliability and precision of highly granular public data for reuse. As inconsistencies might occur in any system, data curators and modelers should better communicate the techniques used to correct unplausible values and thus improve the reproducibility and generalizability of results. We encourage health and media agencies responsible for curating public data to report daily records in a standardized form, avoid using cumulative trend that often masks discrepancies, and provide documentation on the applied corrections of historic records.

We found systematic reporting of 0 counts for federal holidays and weekends throughout our study period. Zero counts corresponded to missing MDPH records. This demonstrates the limitations of cumulative count surveillance reporting: unchanging totals between days will mistake records with missing data as 0 counts. This discrepancy reflects a broader systematic flaw of non-standardized surveillance reporting protocols for infectious disease data repositories [79]. Replacing missing records with 0 counts underreports health outcomes and reduces the precision of near-term forecasts for early outbreak detection. Furthermore, these anomalies introduce greater noise within daily time series that can distort the detection of outbreak events. We implore the scientific community to create standardized protocols and metadata explaining when, why, and how much missing data are present within publicly available records.

Our study did not explore differences in DoW effects according to the introduction and availability of COVID-19 vaccines. We believe that this topic deserves special attention; future research should explore temporal trends and DoW effects for health outcomes including hospitalization, intensive care unit, and vaccination rates. This analysis can utilize the analytic approach demonstrated here to explore segmented temporal trends in DoW effects according to the timing of vaccination rollout by vaccine-accessible subpopulation, type of vaccine, and COVID-19 strain.

In this study we demonstrated applications of adaptive filters as a standardized data-driven approach for defining and comparing outbreak signatures. Public health professionals can apply average smoothers in real-time to monitor the spread, speed, and severity of infectious outbreaks. Rate magnitudes and their trivial derivatives can identify critical points in outbreak signatures so that health professionals might anticipate the acceleration or deceleration in rates of health outcomes. Using these findings, local, state, and national health agencies can identify geographic hotspots of infection, build laboratory testing capacity in locations with emerging outbreak threats, and improve the timeliness of mobilizing medical supplies and personnel to reduce peak rate magnitudes.

## 5. Conclusions

The systematic DoW effects in major outcomes of the COVID-19 pandemic at the different stages of interventions highlight important features of public health responses. The observed patterns across non-pharmaceutical intervention phases reflect the changes in testing capacities, access to healthcare, and preferences imperative for efficiently managing laboratory supplies and public health personnel. Using real-time surveillance and presented methodology, public health agencies can better anticipate testing demands as the COVID-19 pandemic persists. Furthermore, exploration of outbreak signatures can improve the timeliness of mobilizing testing resources and laboratory personnel in anticipation of increased rates of testing and infection. The study also revealed underlying discrepancies in publicly reported health data threatening to underreport SARS-CoV-2 cases and reduce the reproducibility of time series modeling analyses. Public health agencies curating publicly disseminated data must devise universal standards for reporting missing data, correcting historical records, and further improve the reporting infrastructure.

## Figures and Tables

**Figure 1 ijerph-19-01321-f001:**
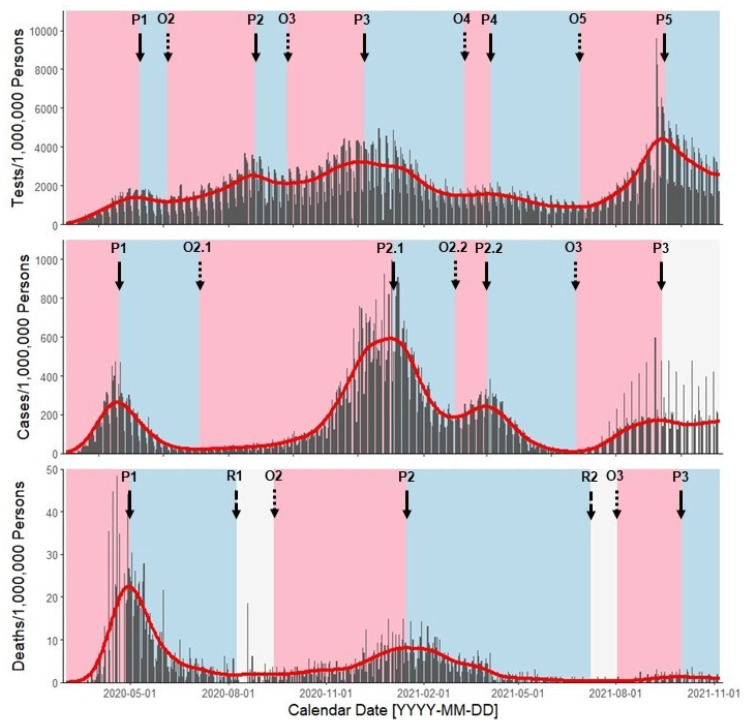
Multi-panel, shared-axis time series plots of MDPH-reported Massachusetts statewide tests (**top** panel) and JHU-reported Middlesex County cases and deaths (**middle** and **bottom** panels, respectively) from 2 March 2020 through 7 November 2021. Within each plot, we provide estimated rates (grey bars) from reported data and 21-day average smoothers (red lines) estimated using Kolmogorov-Zurbenko adaptive filters. Background colors depict acceleration periods (pink), deceleration periods (blue), and steady-state periods (grey) defined independently for each health outcome. We embedded text and arrows at the top of each plot to signify the onset timing (O1–O5, short dashed arrow), peak timing (P1–P5, solid arrow), and resolution timing (R1–R2, long dashed arrow) across outbreak waves. We report rates as tests, cases, and deaths per 1,000,000 persons.

**Figure 2 ijerph-19-01321-f002:**
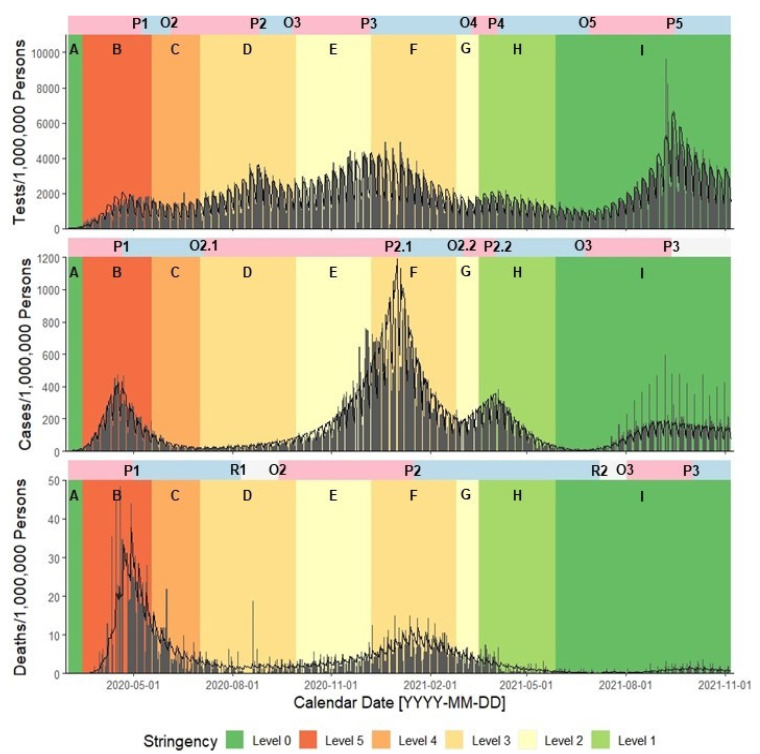
Multi-panel, shared-axis time series plots of MDPH-reported Massachusetts statewide tests (**top** panel) and JHU-reported Middlesex County cases and deaths (**middle** and **bottom** panels, respectively) by outbreak critical period and non-pharmaceutical intervention phase from 2 March 2020 through 7 November 2021. Within each plot, we provide estimated rates (grey bars) from reported data with fitted model results (black lines) from segmented negative binomial regression models adjusted for linear and quadratic trends for each outbreak critical period as defined by Kolmogorov-Zurbenko adaptive filters. Colored bars above each plot depict acceleration (pink), deceleration (blue), and steady-state (grey) periods as reported in Table 3 and Figure 1. We embedded text to signify the onset timing (O1–O5), peak timing (P1–P5), and resolution timing (R1–R2) of critical periods across outbreak waves. Background colors depict phases of Massachusetts non-pharmaceutical interventions from most stringent (red) to least stringent (dark green). We label each intervention phase in accordance with Table 1 (A–I). We report incidence as tests, cases, and deaths per 1,000,000 persons.

**Figure 3 ijerph-19-01321-f003:**
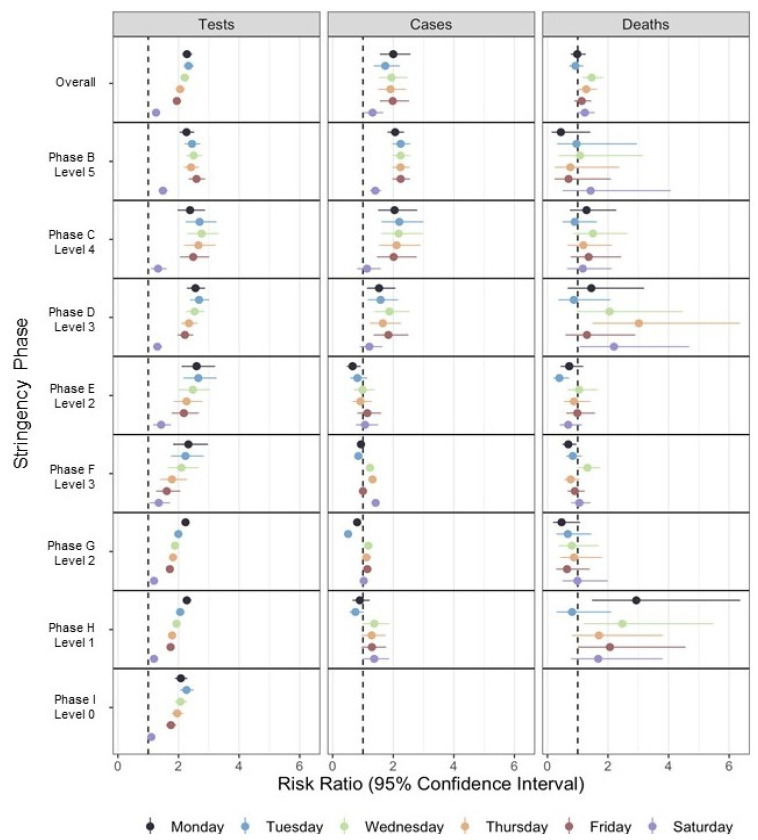
Shared axis, multi-panel forest plot diagrams describing day-of-the-week (DoW) effects for reported tests (**left**), cases (**middle**), and deaths (**right**) by Massachusetts non-pharmaceutical intervention phase. We report DoW effects for the entire study period (Overall, top panels) and by intervention phase from most historic (Phase B) to most recent (Phase I). We used embedded text to list the stringency level (ranging 0–5) for each policy phase. We estimated these effects using segmented negative binomial regression models adjusted for linear and quadratic trends in outbreak critical periods as defined by Kolmogorov-Zurbenko adaptive filters. We denote each day of the week from Monday to Saturday (Reference: Sunday). We report effects as risk ratio (RR) estimates (markers) and 95% confidence intervals (horizontal error bars) denoting RR = 1 with a vertical dashed line across all panels. We report results beginning in Phase B, as this marks the first complete phase aligning with our study period. We exclude Phase I for cases and deaths, as no weekend reporting occurred during this phase.

**Table 1 ijerph-19-01321-t001:** Non-pharmaceutical intervention phases in accordance with Massachusetts statewide COVID-19 health advisories. We defined intervention phases according to the Massachusetts shelter in place order on 15 March 2020 and subsequent reopening of businesses and public venues thereafter [52]. We assigned a stringency score for each phase where 5 reflects the most stringent mandates to reduce social mobility and disease transmission while 0 reflects periods before and after the statewide state of emergency. We report the duration of each phase in days. Phase I duration is reported from the termination of the state of emergency to the end of our study period (7 November 2021).

Phase	Start Date	Duration (Days)	Description	Stringency
A	22 January 2020	53	Period before COVID-19 widespread in the United States	0
B	15 March 2020	64	Closures of public and private elementary and secondary schools; prohibited gatherings of more than 25 people	5
C	18 May 2020	45	Phases I and II reopening	4
D	2 July 2020	89	Phase III-Step 1 reopening	3
E	29 September 2020	70	Phase III-Step 2 reopening	2
F	8 December 2020	79	Return to Phase III-Step 1	3
G	25 February 2021	21	Reentering Phase III-Step 2 reopening	2
H	18 March 2021	71	Phase IV reopening	1
I	28 May 2021	163	Termination of State of Emergency	0

**Table 2 ijerph-19-01321-t002:** Dates of outbreak signature critical points for reported tests in Massachusetts and reported cases and deaths in Middlesex County, Massachusetts, USA from 2 March 2020 through 7 November 2021. We identified critical points using rate magnitudes and trivial derivatives defined using 21-day Kolmogorov-Zurbenko adaptive filter average smoothers. We extracted data from the John Hopkin’s University’s Center for Systems Science and Engineering (CSSE) COVID-19 data repository and Massachusetts Department of Public Health COVID-19 data dashboard. Critical points included the timing of wave onset, the peak of rate magnitudes, and when applicable, the resolution of outbreak waves.

Tests			Cases			Deaths		
Wave	Critical Point	Date	Wave	Critical Point	Date	Wave	Critical Point	Date
1	Onset	2 March 2020	1	Onset	2 March 2020	1	Onset	2 March 2020
1	Peak	8 May 2020	1	Peak	20 April 2020	1	Peak	30 April 2020
2	Onset	4 June 2020	2	Onset 1	4 July 2020	1	Resolution	8 August 2020
2	Peak	25 August 2020	2	Peak 1	2 January 2021	2	Onset	12 September 2020
3	Onset	25 September 2020	2	Onset 2	2 March 2021	2	Peak	15 January 2021
3	Peak	6 December 2020	2	Peak 2	1 April 2021	2	Resolution	7 July 2021
4	Onset	10 March 2021	3	Onset	23 June 2021	3	Onset	1 August 2021
4	Peak	4 April 2021	3	Peak	12 September 2021	3	Peak	1 October 2021
5	Onset	27 June 2021						
5	Peak	15 September 2021						

**Table 3 ijerph-19-01321-t003:** Dates of outbreak signature critical periods for reported tests in Massachusetts and reported cases and deaths in Middlesex County, Massachusetts, USA from 2 March 2020 through 7 November 2021. We identified critical points using rate magnitudes and trivial derivatives defined using 21-day Kolmogorov-Zurbenko adaptive filter average smoothers. We extracted data from the John Hopkin’s University’s Center for Systems Science and Engineering (CSSE) COVID-19 data repository and Massachusetts Department of Public Health COVID-19 data dashboard. We defined critical periods, including acceleration, deceleration, and steady state, as the duration between 2 consecutive critical points. We report the start date for each period, as the end date was defined by the following start date. We list the number of periods defined for each outcome, which were used in segmented time series analyses for modeling day-of-the-week effects.

Tests			Cases			Deaths		
Period	Period Type	Start Date	Period	Critical Point	Start Date	Period	Critical Point	Start Date
1	Acceleration	2 March 2020	1	Acceleration	2 March 2020	1	Acceleration	2 March 2020
2	Deceleration	8 May 2020	2	Deceleration	20 April 2020	2	Deceleration	30 April 2020
3	Acceleration	4 June 2020	3	Acceleration	4 July 2020	3	Steady State	8 August 2020
4	Deceleration	25 August 2020	4	Deceleration	2 January 2021	4	Acceleration	12 September 2020
5	Acceleration	25 September 2020	5	Acceleration	2 March 2021	5	Deceleration	15 January 2021
6	Deceleration	6 December 2020	6	Deceleration	1 April 2021	6	Steady State	7 July 2021
7	Acceleration	10 March 2021	7	Acceleration	23 June 2021	7	Acceleration	1 August 2021
8	Deceleration	4 April 2021	8	Steady State	12 September 2021	8	Deceleration	1 October 2021
9	Acceleration	27 June 2021						
10	Deceleration	15 September 2021						

**Table 4 ijerph-19-01321-t004:** Descriptive summaries of rates for reported tests, cases, and deaths by DoW from 2 March 2020 through 7 November 2021. We extracted data on state-level tests from the Massachusetts Department of Public Health COVID-19 data dashboard and data on county-level cases and deaths from John Hopkin’s University’s COVID-19 data dashboard. We estimated average rates (with 95% confidence intervals) using negative binomial regression models adjusted for log-link functions. We also report median rates (with interquartile range), L-skewness values, and L-kurtosis values by DoW and health outcome.

Day-of-Week	Mean [LCI, UCI]	Median [LQR, UQR]	L-Skewness	L-Kurtosis
Tests				
Sunday	103.28 [83.73, 127.41]	90.89 [63.90, 145.34]	0.16	0.09
Monday	239.26 [194.06, 294.98]	207.89 [146.08, 332.57]	0.14	0.11
Tuesday	248.50 [201.56, 306.37]	195.28 [149.10, 339.75]	0.23	0.17
Wednesday	233.35 [189.27, 287.71]	199.93 [142.05, 307.45]	0.19	0.17
Thursday	213.99 [173.56, 263.85]	183.75 [135.65, 281.46]	0.18	0.17
Friday	199.09 [161.46, 245.48]	179.83 [129.81, 268.36]	0.14	0.13
Saturday	129.26 [104.80, 159.42]	114.61 [78.54, 173.63]	0.13	0.09
Cases				
Sunday	11.69 [9.40, 14.54]	3.85 [0.56, 17.25]	0.47	0.17
Monday	17.06 [13.76, 21.15]	12.69 [3.23, 28.39]	0.25	0.00
Tuesday	14.78 [11.91, 18.34]	12.25 [3.20, 19.33]	0.34	0.17
Wednesday	18.50 [14.93, 22.92]	15.20 [4.25, 24.07]	0.35	0.18
Thursday	18.58 [14.99, 23.02]	14.70 [3.66, 24.23]	0.36	0.19
Friday	17.77 [14.34, 22.03]	13.96 [3.66, 23.92]	0.35	0.19
Saturday	15.45 [12.46, 19.17]	5.96 [1.18, 22.03]	0.49	0.21
Deaths				
Sunday	0.40 [0.27, 0.59]	0.12 [0.00, 0.47]	0.57	0.32
Monday	0.30 [0.20, 0.47]	0.19 [0.06, 0.37]	0.44	0.26
Tuesday	0.32 [0.21, 0.49]	0.12 [0.06, 0.37]	0.53	0.32
Wednesday	0.50 [0.35, 0.72]	0.28 [0.06, 0.59]	0.50	0.32
Thursday	0.37 [0.25, 0.55]	0.25 [0.06, 0.50]	0.43	0.27
Friday	0.36 [0.24, 0.54]	0.19 [0.06, 0.43]	0.46	0.25
Saturday	0.53 [0.37, 0.75]	0.25 [0.00, 0.56]	0.64	0.47

**Table 5 ijerph-19-01321-t005:** Rate ratios (with 95% confidence intervals) of DoW effects relative to Sunday for the full study period (Overall) and by public health policy intervention phase from 2 March 2020 through 7 November 2021. We examined DoW effects using segmented negative binomial regression models adjusted for log-link functions. We adjusted each model with linear and quadratic trends for the outbreak signature critical periods occurring within each intervention phase. We supplement model findings with Akaike’s Information Criterion (AIC) values used to assess model fit. We report results beginning in Phase B, as this marks the first complete phase aligning with our study period. We exclude Phase I for cases and deaths, as no weekend reporting occurred during this phase.

Phase	Monday	Tuesday	Wednesday	Thursday	Friday	Saturday	AIC
Tests							
Overall	2.27 [2.11, 2.45] ^a^	2.33 [2.16, 2.51] ^a^	2.20 [2.05, 2.37] ^a^	2.05 [1.90, 2.21] ^a^	1.94 [1.80, 2.09] ^a^	1.26 [1.17, 1.35] ^a^	9188.30
B	2.26 [2.04, 2.52] ^a^	2.45 [2.20, 2.72] ^a^	2.50 [2.25, 2.78] ^a^	2.41 [2.17, 2.68] ^a^	2.59 [2.34, 2.88] ^a^	1.48 [1.34, 1.65] ^a^	790.47
C	2.45 [2.01, 2.98] ^a^	2.73 [2.24, 3.32] ^a^	2.88 [2.35, 3.53] ^a^	2.72 [2.22, 3.33] ^a^	2.51 [2.05, 3.08] ^a^	1.33 [1.08, 1.63] ^b^	627.75
D	2.56 [2.25, 2.91] ^a^	2.60 [2.28, 2.96] ^a^	2.45 [2.15, 2.79] ^a^	2.30 [2.03, 2.62] ^a^	2.16 [1.90, 2.45] ^a^	1.29 [1.14, 1.47] ^a^	1298.50
E	2.58 [2.12, 3.14] ^a^	2.63 [2.16, 3.21] ^a^	2.46 [2.02, 3.00] ^a^	2.25 [1.84, 2.74] ^a^	2.17 [1.79, 2.65] ^a^	1.43 [1.17, 1.74] ^a^	1107.60
F	2.32 [1.82, 2.97] ^a^	2.23 [1.75, 2.83] ^a^	2.10 [1.65, 2.66] ^a^	1.79 [1.40, 2.28] ^a^	1.61 [1.26, 2.06] ^a^	1.35 [1.05, 1.72] ^c^	1276.10
G	2.24 [2.14, 2.35] ^a^	2.01 [1.92, 2.11] ^a^	1.91 [1.82, 2.01] ^a^	1.83 [1.74, 1.92] ^a^	1.72 [1.63, 1.80] ^a^	1.19 [1.13, 1.25] ^a^	223.09
H	2.27 [2.14, 2.42] ^a^	2.06 [1.93, 2.19] ^a^	1.93 [1.82, 2.06] ^a^	1.79 [1.68, 1.90] ^a^	1.74 [1.63, 1.85] ^a^	1.19 [1.12, 1.27] ^a^	868.16
I	2.07 [1.84, 2.32] ^a^	2.27 [2.02, 2.54] ^a^	2.07 [1.85, 2.33] ^a^	1.96 [1.75, 2.20] ^a^	1.74 [1.55, 1.95] ^a^	1.10 [0.98, 1.23]	2429.90
Cases							
Overall	2.00 [1.56, 2.57] ^a^	1.74 [1.36, 2.22] ^a^	1.94 [1.53, 2.47] ^a^	1.91 [1.50, 2.43] ^a^	1.98 [1.56, 2.53] ^a^	1.32 [1.04, 1.67] ^c^	6783.20
B	2.06 [1.81, 2.35] ^a^	2.24 [1.97, 2.56] ^a^	2.24 [1.97, 2.56] ^a^	2.24 [1.97, 2.55] ^a^	2.25 [1.97, 2.56] ^a^	1.41 [1.23, 1.61] ^a^	587.67
C	2.04 [1.50, 2.79] ^a^	2.20 [1.62, 3.00] ^a^	2.18 [1.60, 2.97] ^a^	2.11 [1.54, 2.90] ^a^	2.01 [1.46, 2.77] ^a^	1.13 [0.81, 1.59]	343.92
D	1.53 [1.13, 2.07] ^b^	1.58 [1.16, 2.16] ^b^	1.87 [1.38, 2.54] ^a^	1.66 [1.22, 2.26] ^b^	1.84 [1.35, 2.50] ^a^	1.21 [0.90, 1.64]	713.90
E	0.66 [0.47, 0.92] ^c^	0.82 [0.58, 1.15]	0.99 [0.70, 1.39]	0.92 [0.65, 1.29]	1.14 [0.81, 1.60]	1.07 [0.76, 1.50]	811.69
F	0.93 [0.89, 0.97] ^b^	0.85 [0.81, 0.88] ^a^	1.23 [1.19, 1.28] ^a^	1.32 [1.27, 1.37] ^a^	1.00 [0.96, 1.04]	1.42 [1.36, 1.47] ^a^	4204.20
G	0.81 [0.71, 0.92] ^b^	0.51 [0.44, 0.59] ^a^	1.17 [1.03, 1.33] ^c^	1.11 [0.97, 1.27]	1.14 [1.02, 1.29] ^c^	1.03 [0.91, 1.15]	365.18
H	0.89 [0.65, 1.22]	0.75 [0.55, 1.03]	1.37 [1.01, 1.87] ^c^	1.29 [0.95, 1.75]	1.30 [0.95, 1.77]	1.37 [1.01, 1.87] ^c^	766.20
Deaths							
Overall	0.98 [0.77, 1.27]	0.92 [0.72, 1.19]	1.46 [1.15, 1.85] ^b^	1.28 [1.00, 1.64] ^c^	1.13 [0.89, 1.45]	1.24 [0.98, 1.57]	2356.60
B	0.44 [0.14, 1.41]	0.96 [0.32, 2.96]	1.08 [0.37, 3.15]	0.76 [0.25, 2.38]	0.70 [0.24, 2.09]	1.43 [0.50, 4.06]	399.09
C	1.29 [0.74, 2.27]	0.90 [0.50, 1.63]	1.50 [0.87, 2.64]	1.18 [0.66, 2.13]	1.36 [0.77, 2.43]	1.17 [0.65, 2.12]	238.51
D	1.45 [0.67, 3.19]	0.87 [0.36, 2.08]	2.05 [0.97, 4.46]	3.02 [1.48, 6.36] ^c^	1.30 [0.59, 2.90]	2.20 [1.06, 4.68] ^c^	342.27
E	0.72 [0.43, 1.18]	0.39 [0.20, 0.72] ^b^	1.04 [0.66, 1.66]	0.88 [0.54, 1.41]	0.99 [0.62, 1.57]	0.69 [0.41, 1.13]	264.52
F	0.69 [0.49, 0.96] ^c^	0.84 [0.61, 1.14]	1.32 [1.00, 1.75] ^c^	0.77 [0.56, 1.06]	0.90 [0.66, 1.23]	1.05 [0.78, 1.42]	401.37
G	0.47 [0.18, 1.08]	0.67 [0.30, 1.45]	0.81 [0.38, 1.70]	0.87 [0.42, 1.80]	0.65 [0.29, 1.39]	0.99 [0.49, 1.99]	92.87
H	2.93 [1.47, 6.37] ^b^	0.81 [0.30, 2.10]	2.47 [1.20, 5.49] ^c^	1.70 [0.81, 3.81]	2.06 [1.00, 4.56]	1.68 [0.78, 3.80]	216.62

Asterisks indicate statistical significance at *p* < 0.001 (^a^), *p* < 0.01 (^b^), and *p* < 0.05 (^c^).

## Data Availability

We provide Excel files with data used to estimate critical points and periods, conduct segmented time series analyses, and create visualizations within our Appendix A. Raw data is publicly accessible on state-level tests from the Massachusetts Department of Public Health COVID-19 data dashboard [42,43] and on county-level cases and deaths from John Hopkin’s University’s COVID-19 data dashboard [37].

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
