# Peer review of "Critical Periods, Critical Time Points and Day-of-the-Week Effects in COVID-19 Surveillance Data: An Example in Middlesex County, Massachusetts, USA"

_ijerph, 2022, doi:10.3390/ijerph19031321_

Round 1

Reviewer 1 Report

In this manuscript by Ryan B. Simpson, et al., have analyzed a few temporal aspects of the COVID-19 pandemic in the Middlesex County, MA.

The authors used public health records and highlighted a few important points both on the pandemic and on public health data quality and consistency making this work relevant not only on understanding the pandemic waves signatures but as well on raising awareness on public health database consistency.

The problems raised by the authors are common with manual data entry and can be detrimental for surveillance studies as well for public perception and deserve more attention. On the COVID-19 pandemic wave signatures analysis, the authors were successful in supporting their main points and the analysis methodology is sound.

However, a few minor points need to be addressed. I've numbered my main issues and questions.

  1. The authors calculated the L-skewness and L-kurtosis and presented the results in a table but didn`t really discuss their significance. And what these numbers represent in the context of the data presented. That would be a great addition to the discussion.
  2. More in the statistical results summarized on the table 4 are hard to interpret. Maybe it would be helpful to represent the result in a graph.
  3. Seems like the author switched the colors in the description of Figure 1. In addition, it would probably be beneficial to change the colors a little bit to improve contrast, more specifically between the blue and gray areas. The orange looks more like pink and the blue is very similar to the gray being easy to get them confused.
  4.  “We identified a global maximum of cases for the primary peak of Wave 2 (02 January 2021; 591.37cpm)…” [line 288]; “Wave 3 onset timing (25 September 2021) aligned with the relaxation to less-stringent Phase E (29 September 2020) while Wave 3 peak timing (06 December 2020) aligned with the return…” [line 358]; “Alternatively, increased testing may reflect premature relaxation in the stringency of non-pharmaceutical intervention phases. We found that the onset timing of reported cases in Wave 2 (04 July 2020)…” [line 366]; And may others. Better highlight the periods discussed on the plots. In general, for figures 2 and 3, it is important to better highlight key points discussed in the text. In the text, the authors frequently reference important dates that are not directly represented in the X axis. It is very hard to go along the text when the most important dates are not represented in the scale or anywhere in the pots. I suggest either changing the X axis tick marks or somehow highlight the important dates in the figure (maybe arrows).
  5. Although it is not in the overall scope of the work, with the introduction and availability of the vaccines in 2021, there was a big shift in the public perception of the pandemic and heath risks. Did the authors investigate the effect of the vaccination?
  6. For figure 3, it would be helpful to add stringency information on the plot itself. The reader must scroll back making it a little distracting.
  7. What does the longer confidence intervals, mainly in the phases D and H mean for deaths? Is it a data collection problem (may inconsistency in data entry) or an effect of the pandemic state ate these times?
  8. “Overall, we found minimal or no significant differences in reported deaths between any days of the week, especially in the most stringent Phases B-C.” [line 333]; “Given such characterization, we identified significant DoW effects for reported health outcomes with distinct patterns across intervention phases.” [line 345]. The authors point out that minimal or no significant differences in reported deaths were observed between any days of the week, however, shortly after, the authors state that they identified significant DoW effects for reported health outcomes with distinct patterns across intervention phases (examples on lines 333 and 345). This seems a little inconsistent. To me, it seems that the DoW effect on death is really weak, which is not really surprising. It seems like the really interesting differences are between the intervention phases. Since this is one of the main points proposed, the authors must make their interpretation more clear.
  9. The combination of these findings indicates health professionals must place greater attention to investigating DoW effects and other reporting inconsistencies in public surveillance data. [line 354]. This is an important point raised by the authors. Inconsistency in data entry is really common, especially in manually entered information. This is not an issue commonly discussed and clearly deserves more attention.
  10. It is not clear to me if, during the initial data cleanup and preprocessing, if the reported case is retrofitted to test date. Since the test results would not be released before 24h or 48h, one would expect that testing on weekends, due to reduced capacity would serve mostly people already suspected exposure so, I`d expect an increased positivity rate for these tests. Could this analysis be done with the present data? (number of cases in relation to number of tests).
  11. “ Assuming a 5-7-day incubation period before presenting with COVID-19 symptoms, it is also likely that the changes in DoW effects in part reflect changes in transmission and social mixing when stay-at-home policies and business and retailer closures were relaxed in 2021.” [line 397]. This could be true, but with the presented data, it is hard to make this conclusion. The DoW doesn`t seem to be too significant, and also, due to, I would expect, a much looser correlation between exposure and test (the 5 – 7 incubation days would be much more spread (people would test before the 5 days when known to have made contact with confirmed cases, and would test later than the 7 days, due to difficulties in scheduling tests or as personal choice. Maybe this could be back traced with “Days post symptom onset”, which would be a more precise measure. Again, authors must make clearer their interpretation of the DoW effects on their variables.
  12.  “We expected to find no DoW effects for deaths, as the likelihood of death is equally probable on any day of the week.” [line 400]. In retrospect yes, but prior to the analysis you cant really be sure. If there was a strong DoW effect on cases, a stronger effect on the DoW for deaths could be expected (although this would need to be corrected for other confounding variables like age, overall health, BMI, etc.. ).

Author Response

In this manuscript by Ryan B. Simpson, et al., have analyzed a few temporal aspects of the COVID-19 pandemic in the Middlesex County, MA.

The authors used public health records and highlighted a few important points both on the pandemic and on public health data quality and consistency making this work relevant not only on understanding the pandemic waves signatures but as well on raising awareness on public health database consistency.

The problems raised by the authors are common with manual data entry and can be detrimental for surveillance studies as well for public perception and deserve more attention. On the COVID-19 pandemic wave signatures analysis, the authors were successful in supporting their main points and the analysis methodology is sound.

We thank the Reviewer.

However, a few minor points need to be addressed. I've numbered my main issues and questions.

The authors calculated the L-skewness and L-kurtosis and presented the results in a table but didn`t really discuss their significance. And what these numbers represent in the context of the data presented. That would be a great addition to the discussion.

We have modified the Methods:

“We selected L-skewness and L-kurtosis given their sensitivity for distributions of low rates and smaller sample size [53]. High values of L-moments indicate a possible surge in tests, cases, or deaths and are necessary for justifying the use of negative binomial distributions when modeling count-based distributions. We estimated average rates using generalized linear models adjusted for a negative binomial distribution and log-link function…”

Please note that we interpret the significance of L-moments in the Results:

“As expected for each death there were about 30 cases of infections, and for each reported case there were about 10 reported tests. Median rates were high on weekdays for tests and cases yet low on weekends; we found little differences between mean or median rates of deaths across all days of the week. Relatively high values of L-skewness and L-kurtosis for daily death cases suggested the potential for high temporal variability.

We have modified the Discussion:

“We expected to find no DoW effects for deaths, as the likelihood of death is equally probable on any day of the week. Yet, we found several spikes in death counts we can’t explain; those spikes triggered occasional significant effects. These spikes are supported by high values of L-skewness and L-kurtosis suggesting temporal variability of deaths within our time series. Future studies could also examine compound effects of additional contributors, like local air quality…”

More in the statistical results summarized on the table 4 are hard to interpret. Maybe it would be helpful to represent the result in a graph.

We have modified the Results:

“As expected for each death there were about 30 cases of infections, and for each reported case there were about 10 reported tests. Median rates were high on weekdays for tests and cases yet low on weekends; we found little differences between mean or median rates of deaths across all days of the week. Relatively high values of L-skewness and L-kurtosis for daily death cases suggested the potential for high temporal variability.”

Seems like the author switched the colors in the description of Figure 1. In addition, it would probably be beneficial to change the colors a little bit to improve contrast, more specifically between the blue and gray areas. The orange looks more like pink and the blue is very similar to the gray being easy to get them confused.

We thank the reviewer for identifying the switching of colors in figure legends. The legends for Figure 1 and 2 have been corrected. We have revised Figure 1 and 2 background colors for acceleration and deceleration periods to include darker background colors (pink and blue, respectively) that improve contrast.

“We identified a global maximum of cases for the primary peak of Wave 2 (02 January 2021; 591.37cpm)…” [line 288]; “Wave 3 onset timing (25 September 2021) aligned with the relaxation to less-stringent Phase E (29 September 2020) while Wave 3 peak timing (06 December 2020) aligned with the return…” [line 358]; “Alternatively, increased testing may reflect premature relaxation in the stringency of non-pharmaceutical intervention phases. We found that the onset timing of reported cases in Wave 2 (04 July 2020)…” [line 366]; And may others. Better highlight the periods discussed on the plots. In general, for figures 2 and 3, it is important to better highlight key points discussed in the text. In the text, the authors frequently reference important dates that are not directly represented in the X axis. It is very hard to go along the text when the most important dates are not represented in the scale or anywhere in the pots. I suggest either changing the X axis tick marks or somehow highlight the important dates in the figure (maybe arrows).

To improve the clarity of identifying critical periods and policy phases, we have superimposed embedded text onto Figures 1 and 2. This includes arrows identifying each critical period onset, peak, and resolution as well as labels for policy phases A-I.

Although it is not in the overall scope of the work, with the introduction and availability of the vaccines in 2021, there was a big shift in the public perception of the pandemic and heath risks. Did the authors investigate the effect of the vaccination?

We thank the Reviewer for this excellent point. While the introduction and availability of vaccines is an important event to consider, we believe that this topic deserves special attention in a new manuscript. Future research could explore temporal trends and DoW effects before and after introduction of vaccination doses and its availability to different subpopulations. Such an analysis would require examination of the proportion of population vaccinated as well as hospitalization and intensive care unit rates.

We have modified the Discussion to comment on the need for future research:

Our study did not explore differences in DoW effects according to the introduction and availability of COVID-19 vaccines. We believe that this topic deserves special attention; future research should explore temporal trends and DoW effects for health outcomes including hospitalization, intensive care unit, and vaccination rates. This analysis can utilize the analytic approach demonstrated here to explore segmented temporal trends in DoW effects according to the timing of vaccination rollout by vaccine-accessible subpopulation, type of vaccine, and COVID-19 strain.

For figure 3, it would be helpful to add stringency information on the plot itself. The reader must scroll back making it a little distracting.

We have revised Figure 3 to include information on the intervention phase and stringency level in vertical axis labels.

What does the longer confidence intervals, mainly in the phases D and H mean for deaths? Is it a data collection problem (may inconsistency in data entry) or an effect of the pandemic state ate these times?

We have modified the Results:

“Overall, we found minimal or no significant differences in reported deaths between any days of the week, especially in the most stringent Phases B-C. As later intervention phases had many weeks without reported deaths, confidence intervals were exaggerated in regression analyses. The observed increase in reported deaths on Wednesdays and Thursdays (1.46-times [1.15, 1.85], p=0.002) and 1.28-times ([1.00, 1.64], p=0.047, respectively) were driven by spikes in deaths in individual critical periods and intervention phases.”

“Overall, we found minimal or no significant differences in reported deaths between any days of the week, especially in the most stringent Phases B-C.” [line 333]; “Given such characterization, we identified significant DoW effects for reported health outcomes with distinct patterns across intervention phases.” [line 345]. The authors point out that minimal or no significant differences in reported deaths were observed between any days of the week, however, shortly after, the authors state that they identified significant DoW effects for reported health outcomes with distinct patterns across intervention phases (examples on lines 333 and 345). This seems a little inconsistent. To me, it seems that the DoW effect on death is really weak, which is not really surprising. It seems like the really interesting differences are between the intervention phases. Since this is one of the main points proposed, the authors must make their interpretation more clear.

We have modified the Results:

“We found patterns in the timing of wave onset, peak, and resolution critical points compared to non-pharmaceutical intervention phases. Given such characterization, we identified significant DoW effects for Massachusetts statewide tests and Middlesex County cases with distinct patterns across intervention phases. We also identified and reconciled systematic reporting inconsistencies between JHU and MDPH surveillance systems, including dissimilar missing data reporting protocols and the reporting of negative counts to correct historic records.”

The combination of these findings indicates health professionals must place greater attention to investigating DoW effects and other reporting inconsistencies in public surveillance data. [line 354]. This is an important point raised by the authors. Inconsistency in data entry is really common, especially in manually entered information. This is not an issue commonly discussed and clearly deserves more attention.

We thank and agree with the Reviewer that questions of data quality and reliability are imperative to address within COVID-19 modeling research.

It is not clear to me if, during the initial data cleanup and preprocessing, if the reported case is retrofitted to test date. Since the test results would not be released before 24h or 48h, one would expect that testing on weekends, due to reduced capacity would serve mostly people already suspected exposure so, I`d expect an increased positivity rate for these tests. Could this analysis be done with the present data? (number of cases in relation to number of tests).

We thank the Reviewer for this excellent point and agree that it requires additional attention. Unfortunately, our dataset using state-level tests and county-level cases, preventing the use of their ratio to reliably estimate positivity rates. However, we acknowledge that such a study is imperative to more closely track how cases and tests are processed at the county level. Such an analysis would require a differing study design using contact tracing data to gauge when persons become tested, positivity rates of reported tests, and possible symptomology of test seekers. Such a study design would be expensive, and unfortunately the granularity of publicly reported surveillance data does not support these types of analyses.

“ Assuming a 5-7-day incubation period before presenting with COVID-19 symptoms, it is also likely that the changes in DoW effects in part reflect changes in transmission and social mixing when stay-at-home policies and business and retailer closures were relaxed in 2021.” [line 397]. This could be true, but with the presented data, it is hard to make this conclusion. The DoW doesn`t seem to be too significant, and also, due to, I would expect, a much looser correlation between exposure and test (the 5 – 7 incubation days would be much more spread (people would test before the 5 days when known to have made contact with confirmed cases, and would test later than the 7 days, due to difficulties in scheduling tests or as personal choice. Maybe this could be back traced with “Days post symptom onset”, which would be a more precise measure. Again, authors must make clearer their interpretation of the DoW effects on their variables.

We have modified the Results:

“Changes in weekday DoW effects for reported cases might reflect changes in hospitalization patterns, severity of infections, and potential delays due to high testing volumes earlier in the week or during wave peaks. DoW effects for tests and cases may also reflect changes in transmission and social mixing when stay-at-home policies and business and retailer closures were relaxed in 2021. Alternatively, we found no DoW effects for deaths, as the likelihood of death was equally probable on any day of the week. Yet, we found several spikes in death counts we can’t explain; those spikes triggered occasional significant effects.”

“We expected to find no DoW effects for deaths, as the likelihood of death is equally probable on any day of the week.” [line 400]. In retrospect yes, but prior to the analysis you cant really be sure. If there was a strong DoW effect on cases, a stronger effect on the DoW for deaths could be expected (although this would need to be corrected for other confounding variables like age, overall health, BMI, etc.. ).

We have modified the Results:

“Future studies must examine compound effects of additional contributors, like local air quality [76], extreme weather [77], specific calendar events (see Supplementary Table S1), an introduction of rapid over-the-counter tests [11] influencing viral transmission and thus, daily COVID-19 outcomes. Studies may also consider more exhaustive adjustment of confounding factors expected to influence case or death onset such as age, chronic health conditions, body mass index, etc.

Reviewer 2 Report

the article is innovative and interesting and covers fundamental aspects of the pandemic; however, some things are not clear: 
1) have the guidelines adopted worldwide been taken into account, i.e. WHO, ECDC, Italian ministry (Jordana J, Triviño-Salazar JC. Where are the ECDC and the EU-wide responses in the COVID-19 pandemic? Lancet. 2020 May 23;395(10237):1611-1612. doi: 10.1016/S0140-6736(20)31132-6. Epub 2020 May 13. PMID: 32410757; PMCID: PMC7220161.Gherlone E, Polizzi E, Tetè G, Capparè P. Dentistry and Covid-19 pandemic: operative indications post-lockdown. New Microbiol. 2021 Jan;44(1):1-11. Epub 2020 Oct 31. PMID: 33135082.), and whether or not they can influence the course of contagions?
2) I do not understand the method used, especially in the diversity compared to the ministerial monitoring agencies, i.e. why should they use this method rather than their own? clarify it better
3) have vaccines been considered?
4) is it correct to link deaths to tests? does it not also depend on the facilities available, the doctors, the staff, the hospiatl's beds and the demographic characteristics?

Author Response

The article is innovative and interesting and covers fundamental aspects of the pandemic; however, some things are not clear: 

Have the guidelines adopted worldwide been taken into account, i.e. WHO, ECDC, Italian ministry (Jordana J, Triviño-Salazar JC. Where are the ECDC and the EU-wide responses in the COVID-19 pandemic? Lancet. 2020 May 23;395(10237):1611-1612. doi: 10.1016/S0140-6736(20)31132-6. Epub 2020 May 13. PMID: 32410757; PMCID: PMC7220161.Gherlone E, Polizzi E, Tetè G, Capparè P. Dentistry and Covid-19 pandemic: operative indications post-lockdown. New Microbiol. 2021 Jan;44(1):1-11. Epub 2020 Oct 31. PMID: 33135082.), and whether or not they can influence the course of contagions?

We thank the Reviewer for providing these interesting and insightful references. However, our study focuses only on the narrow state-level and county-level tests, cases, and deaths with Massachusetts and Middlesex County, respectively. While international guidelines may be interesting to consider, we limited our scope to policies applicable within these geographic locations, thereby creating the public health policy intervention phases found in Table 1 and Supplementary Table S1.

I do not understand the method used, especially in the diversity compared to the ministerial monitoring agencies, i.e. why should they use this method rather than their own? clarify it better

Unfortunately, we are unable to understand the question asked by the Reviewer. Our manuscript does not speak to questions of diversity nor ministerial monitoring agencies or their methods for modeling COVID-19. Our analytic approach aimed at applying Kolmogorov Zurbenko adaptive filters, well established as a filtering method, to COVID-19 daily time series data to define outbreak signatures. We compared features of outbreak features, namely onset, peak, and resolution timing, to the transition of policy intervention phases as defined by local and statewide public health agencies. This timeline of events was curated by the Massachusetts Department of Public Health and verified for Middlesex County by numerous publicly available sources cited in our Supplementary Materials. Finally, day-of-the-week or DOW effects were modeled by comparing the rates of tests, cases, and deaths of all weekdays and Saturday to Sundays. This analysis was warranted given the rarity by which it is performed on public health surveillance data and the temporal granularity of daily records.

Have vaccines been considered?

We thank the Reviewer for this excellent point. While the introduction and availability of vaccines is an important event to consider, we believe that this topic deserves special attention in a new manuscript. Future research could explore temporal trends and DoW effects before and after introduction of vaccination doses and its availability to different subpopulations. Such an analysis would require examination of the proportion of population vaccinated as well as hospitalization and intensive care unit rates.

We have modified the Discussion to comment on the need for future research:

Our study did not explore differences in DoW effects according to the introduction and availability of COVID-19 vaccines. We believe that this topic deserves special attention; future research should explore temporal trends and DoW effects for health outcomes including hospitalization, intensive care unit, and vaccination rates. This analysis can utilize the analytic approach demonstrated here to explore segmented temporal trends in DoW effects according to the timing of vaccination rollout by vaccine-accessible subpopulation, type of vaccine, and COVID-19 strain.

Is it correct to link deaths to tests? does it not also depend on the facilities available, the doctors, the staff, the hospiatl's beds and the demographic characteristics?

We believe it is imperative to link tests and deaths within our manuscript. Using the Kolmogorov Zurbenko filter, we defined and systematically compared outbreak signatures between all health outcomes. These differences, in combination with their relationship to health policy intervention phases, allows for tracking major pandemic surges in near-real time, understanding how to best utilize laboratory capacity, and begin to question popular preferences in testing. Furthermore, comparisons between tests and deaths helps to raise questions on how to build capacity when monitoring population-level outbreaks including both minimizing the volume and severity of infected persons. By demonstrating the utility of publicly available data, we demonstrate the utility of daily, county-level surveillance data collection and reporting. This can ensure greater preparedness and attention to establishing surveillance systems in the infancy of pandemic outbreaks.